

# Evaluating climate change impacts on streamflow variability based on a multisite multivariate GCM downscaling method

Zhi Li[1], Jiming Jin[2,3]

[1]College of Natural Resources and Environment, Northwest A & F University, Yangling, Shaanxi, 712100, China
[2]College of Water Resources and Architectural Engineering, Northwest A&F University, Yangling, Shaanxi, 712100, China
[3]Departments of Watershed Sciences, Utah State University, Logan, UT, USA

*Correspondence to*: Zhi Li (lizhibox@nwafu.edu.cn)

**Abstract.** Projected hydrological variability is important to future water resources management. Such a projection is often driven by downscaled general circulation model variables. This study developed the multisite multivariate climate change scenarios through three steps: (i) spatially downscaling GCMs with a transfer function method, (ii) temporally downscaling GCMs with a single-site weather generator, and (iii) reconstructing the spatiotemporal correlations with a nonparametric shuffle procedure. Through these steps, multisite precipitation and temperature change scenarios for the period of 2011–2040 were generated from five GCMs under four Representative Concentration Pathways (RCP) to project future changes in streamflow variability with the Soil and Water Assessment Tool (SWAT) for the Jing River catchment on China's Loess Plateau. The correlation reconstruction method performed well for inter-site and -variable correlation reproduction and hydrological modeling. SWAT model was well calibrated with monthly streamflow to have a model efficiency coefficient of 0.78. The annual mean precipitation would not change, while the mean maximum and minimum temperatures would significantly increase by 1.6±0.3 and 1.3±0.2 ℃; the variance ratios of 2011–2040 to 1961–2005 were 1.15±0.13 for precipitation, 1.15±0.14 for mean maximum temperature and 1.04±0.10 for mean minimum temperature. A warmer climate was detected for the flood season while winter and spring would be wetter and warmer; the intra- and inter-annual variations of climate would be greater than those of the current climate. The total annual streamflow would change insignificantly, but its variance ratios of 2011–2040 to 1961-2005 would increase by 1.25±0.55. The streamflow variability would be greater over most months at a seasonal scale due to the increase of monthly maximum streamflow and the decrease of monthly minimum streamflow. The increase in streamflow variability was mainly attributed to the larger positive contributions from increased precipitation variances than the negative contributions from increased temperature means.

## 1 Introduction

Hydrological variability, other than changes in mean state, can cause more disasters such as flooding or drought, and seriously threaten the natural and social systems. The worldwide detected changes in hydrological variability have been paid a great attention in recent years (Chen et al., 2014; Chevalier et al., 2014; Rudorff et al., 2014; Tarhule et al., 2015); especially, the potential changes in hydrological variability under future climate change have been largely evaluated to





provide information for water resources management (Zierl and Bugmann, 2005; Hirabayashi et al., 2013; Arnell and Gosling, 2014; Dankers et al., 2014; Prudhomme et al., 2014; Schewe et al., 2014). However, the response of hydrological variability to climate variability has not been conclusive due to limited surface observations (IPCC, 2013), complex watershed properties (Andrés-Doménech et al., 2015), hydrological modeling and development of climate change scenarios (Wilby and Harris, 2006). The extent to which the hydrological variability is influenced by climate variability should be more thoroughly investigated based on technique improvement.

According to the global-scale projections, the hydrological variability will not change uniformly across the globe. For example, the 30-yr flood will occur more frequently over 50% of the globe (Dankers et al., 2014) and the increase in hydrological drought are projected for over 40% of analyzed land area (Prudhomme et al., 2014). Besides catchment properties, the spatial variations in hydrological variability changes are due to those in climate changes. For example, the drought increase is generally located where precipitation decreases; however, drought still increases in some areas with increased precipitation if stronger evaporation is driven by temperature increase (Prudhomme et al., 2014). Thus, the mechanism by which climate variability influences hydrological variability should be analyzed.

The largest uncertainties in impact assessment of climate change on hydrology originate from climate change scenario development, including GCMs, emission scenarios, and downscaling methods (Wilby and Harris, 2006; Kingston and Taylor, 2010; Chen et al., 2011). These uncertainties, to some extent, can be interpreted by introducing multiple climate models, emission scenarios, and downscaling methods. However, one aspect related to downscaling method, i.e. reconstructing spatial structure of climate, has been paid a great attention in downscaling technique, but rarely been incorporated in hydrological modeling. When hydrological models are applied to large-scale catchments with multisite climate not considering spatial structure, the high flow in one sub-basin can be offset by low flow in a neighboring sub-basin (Wilks, 1998; Thyer and Kuczera, 2003; Clark et al., 2004b; Wheater et al., 2005). Therefore, failure to feed hydrological models with spatiotemporally correlated climate would reduce hydrological variability and potentially misrepresent climate risks.

Numerous multisite downscaling methods have been developed, such as dynamic methods based on regional climate models (RCMs) (Cooley and Sain, 2010; Bárdossy and Pegram, 2012; Pegram and Bárdossy, 2013), empirical scaling methods (Allerup, 1996; Bürger and Chen, 2005), Generalized Linear Models (GLM) (Wheater et al., 2005; Yang et al., 2005; Lu and Qin, 2014; Asong et al., 2016), Artificial Neural Network (ANN) (Harpham and Wilby, 2005; Cannon, 2008), Nonhomogeneous Hidden Markov Model (NHMM) (Charles et al., 1999; Bellone et al., 2000; Fu et al., 2013), and weather generators (Wilks, 1999a; Qian et al., 2002; Mehrotra and Sharma, 2010; Khalili et al., 2013; Srivastav and Simonovic, 2015). So far, their application to hydrological modeling is limited for most methods except for the stochastic weather generator. For example, the parametric weather generators were used to investigate the response of hydrological variability to the spatial structure of climate (Watson et al., 2005; Khalili et al., 2011; Chen et al., 2016; Li et al., 2017), and generate climate change scenarios to assess the changes in irrigated agriculture in Chile (Meza et al., 2012).



The parametric weather generators, including the Richardson-type model and circulation-based model (Katz and Parlange, 1996), are widely used for hydrological modeling because of easy implementation. The Richardson-type model directly perturbs the daily weather generator parameters based on changes in the corresponding monthly statistics (Wilks, 1999b), and the circulation-based model specifies daily-varying parameters using regressions between local predictands and large-scale predictors (Wilby et al., 2002). After parametric adjustment, the two models are driven with correlated random numbers to capture the spatial structure exhibited in the daily weather data. Especially, with direct parametric adjustments, the Richardson-type model could be used more fruitfully either for impact assessment or sensitivity analysis. Specifically, it can be used in the 'top down' framework of impact study by downscaling GCM outputs to assess the potential impacts, and also the 'bottom-up' or stress-testing adaptation options to explore the sensitivity of hydrology to changed climate conditions by direct adjustments of daily weather generator parameters.

For the Richardson-type weather generators, the generation of correlated random numbers is computationally-intensive. Instead, an improvement of the Richardson-type model, through replacing the preprocessing steps of random number generation with a postprocessing procedure for recorrelating the generated data, has been elaborated in recent years because of its high efficiency and good performance. Currently, the algorithm improvement is only carried out for multisite simulation of precipitation without consideration of multivariate correlation (Tarpanelli et al., 2012; Li, 2014). Further extension of this method to multisite and multivariate downscaling will promote the application of weather generator for impact assessment of climate change.

Considering the importance of evaluating the potential changes in hydrological variability and the difficulty in applying the Richardson-type weather generator to multisite and multivariate downscaling, this study is to extend an efficient multisite precipitation generator, i.e. two-stage weather generator (TSWG) (Li, 2014), to a multisite and multivariate GCM downscaling method, and to further assess the impacts of climate change on streamflow variability in a river basin on China's Loess Plateau by combining the generated climate change scenarios with a distributed hydrological model. This study presents a framework for maximizing the application of the parametric weather generator to multisite and multivariate simulation as well as provides useful information for water resources management.

## 2 Data and Methodology

### 2.1 Data description

To project the impact of climate change on hydrology, two datasets are essential: one for climate change scenario development and the other for hydrological simulation. For climate change scenario development, climate forcing data included historical daily precipitation (P) and maximum and minimum temperatures ($T_{max}$ and $T_{min}$) from 18 meteorological stations for the period of 1961–2005, and GCM-simulated and projected monthly P, $T_{max}$, and $T_{min}$ for the period of 1961–



2005 and 2011–2040, respectively. The period 2011–2040 was chosen for impact study because of two reasons: (i) the results from the near-term horizon can be directly used for adaption, and (ii) the uncertainties in climate projection are the least since they are increasing over time due to the uncertainties from GCMs and emission scenarios (IPCC, 2013).

**Table 1.** Data used for future climate change scenario construction

| GCM | Institute | Resolution | Emission scenarios[*] |
|---|---|---|---|
| CanESM2 | Canadian Centre for Climate Modelling and Analysis (Canada) | 2.8 °×2.8 ° | his, rcp2.6, rcp4.5, rcp8.5 |
| CSIRO-Mk3.6.0 | Commonwealth Scientific and Industrial Research Organization in collaboration with Queensland Climate Change Centre of Excellence (Australia) | 1.875 °×1.875 ° | his, rcp2.6, rcp4.5, rcp6.0, rcp8.5 |
| GFDL-CM3 | Geophysical Fluid Dynamics Laboratory (USA) | 2.5 °×2.0 ° | his, rcp2.6, rcp4.5, rcp6.0, rcp8.5 |
| HadGEM2-ES | Met Office Hadley Centre (UK) | 1.875 °×1.25 ° | his, rcp2.6, rcp4.5, rcp6.0, rcp8.5 |
| MPI-ESM-LR | Max Planck Institute for Meteorology (Germany) | 1.875 °×1.875 ° | his, rcp2.6, rcp4.5, rcp8.5 |

Five GCMs (CanESM2, CSIRO-Mk3.6.0, GFDL-CM3, HadGEM2-ES, and MPI-ESM-LR) under historical and four

representative concentration pathways (RCP2.6, 4.5, 6.0, and 8.5) emission scenarios from Intergovernmental Panel on Climate Change – the fifth Assessment Report (IPCC AR5) were used. The GCMs provide data for almost all emission scenarios except for CanESM2 and MPI-ESM-LR that have no data for the RCP 6.0 (Table 1). The four RCPs are named for the radiative forcing values for the year 2100 (2.6, 4.5, 6.0, and 8.5 W/m$^2$, respectively). These RCPs cover most possible future greenhouse emission scenarios, and GCMs associated with these RCPs have projected significant temperature rises

(IPCC, 2013).

The dataset for hydrological simulation is from the Jing River catchment on China's Loess Plateau of China (Figure 1). As SWAT (Soil and Water Assessment Tool) (Arnold et al., 1998) is used to simulate the hydrological cycle, data and/or maps related to climate, soil, vegetation, and hydrology are essential. These datasets were collected from the Data Sharing Infrastructure of Loess Plateau, including daily weather data from 18 stations, soil map and properties, land use map for

1986, monthly streamflow at the catchment outlet (the Zhangjiashan station).

The Jing River catchment is selected as the study area because it is a typical catchment with high intra-annual and inter-annual variability of climate and runoff. The variability has been threatening the water resources management and soil erosion. The catchment has an area of 45,421 km$^2$ and is located within a transition zone between a subhumid and semiarid climate. Although the area-averaged annual precipitation was only 542.1 mm, 55% of the precipitation fell in the flood

season between July and September (1961-2005). The several extreme rainfall events can generate severe soil erosion with soil loss of 5,015 tons km$^{-2}$ a$^{-1}$ over 1961–2000. In addition, with a dry climate and runoff ratio of 7%, the catchment is subject to a severe water shortage. Obviously, the water-related problems in the Jing River catchment are highly correlated





with climatic and hydrological variability. Therefore, potential impacts of climate changes on hydrological variability should be further evaluated.

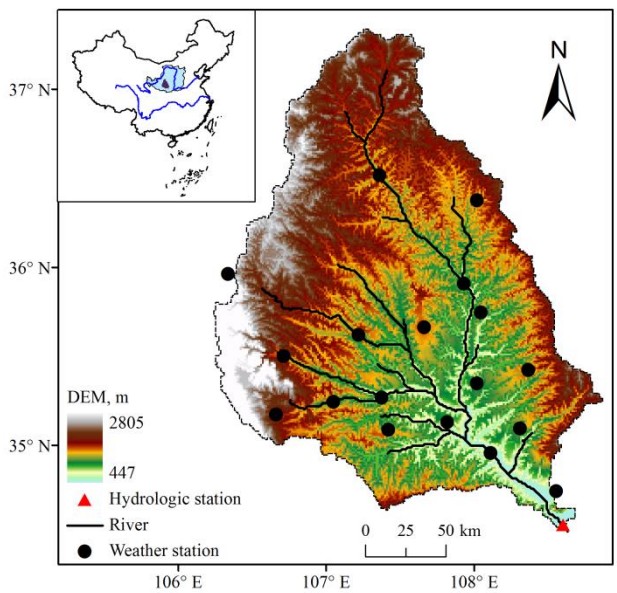

Figure 1. Location of the Jing River catchment

## 2.2 Multisite and multivariate downscaling

The multisite and multivariate GCM downscaling was carried out through three steps. The first step spatially downscaled GCM outputs from a grid scale to a station scale, the second step disaggregated the spatially downscaled GCM outputs from a monthly scale to a daily scale, and the third step reconstructed the multisite and multivariate correlations. The schematic diagram of the methodology is presented in Figure2 and a brief introduction of the methods is given as follows.

The first and second steps are for single-site GCM downscaling, and the popular technique is to combine transfer function method for spatial downscaling with weather generator for temporal downscaling. Here, we used parametric quantile mapping method and a Richardson-type weather generator for these two steps (Zhang and Liu, 2005; Li et al., 2011). For the first step, a linear and nonlinear transfer functions were respectively fitted with the rank-ordered monthly observations and GCM data for each calendar month over 1961–2005, and then applied to the period of 2011–2040 to calculate the monthly mean and variance. The nonlinear function was used to transform the GCM monthly precipitation values that were within the range in which the nonlinear function was fitted, while the linear function was used for the values outside the range. Temporal downscaling was then implemented by adjusting the precipitation- and temperature-related parameters of a single-site weather generator (SSWG) calculated from the baseline period. In SSWG, the precipitation occurrence and amount were simulated by the first-order two-state Markov Chain and the skewed normal distribution based on our previous evaluation





(Li et al., 2014), while temperature was generated by the normal distribution. Thus, the precipitation-related and temperature-related parameters for adjustment include transitional probabilities of wet day following wet day ($P_{w/w}$) and wet day following dry day ($P_{w/d}$), mean and variance of daily precipitation of wet days and mean and variance of the maximum and minimum temperature. They were adjusted according to some relationships developed with the observation. The detailed procedure can be found in Zhang and Liu (2005) and Li et al. (2011). The adjusted parameters were used to drive SSWG to obtain climate change scenarios for a period of 100 years. To test the performance of our method for reconstruction of multisite and multivariate correlation, the temperature generation was not dependent on the dry/wet status as the regular routine in weather generators, and no inter-site or inter-variable correlations were taken into account during the single-site downscaling.

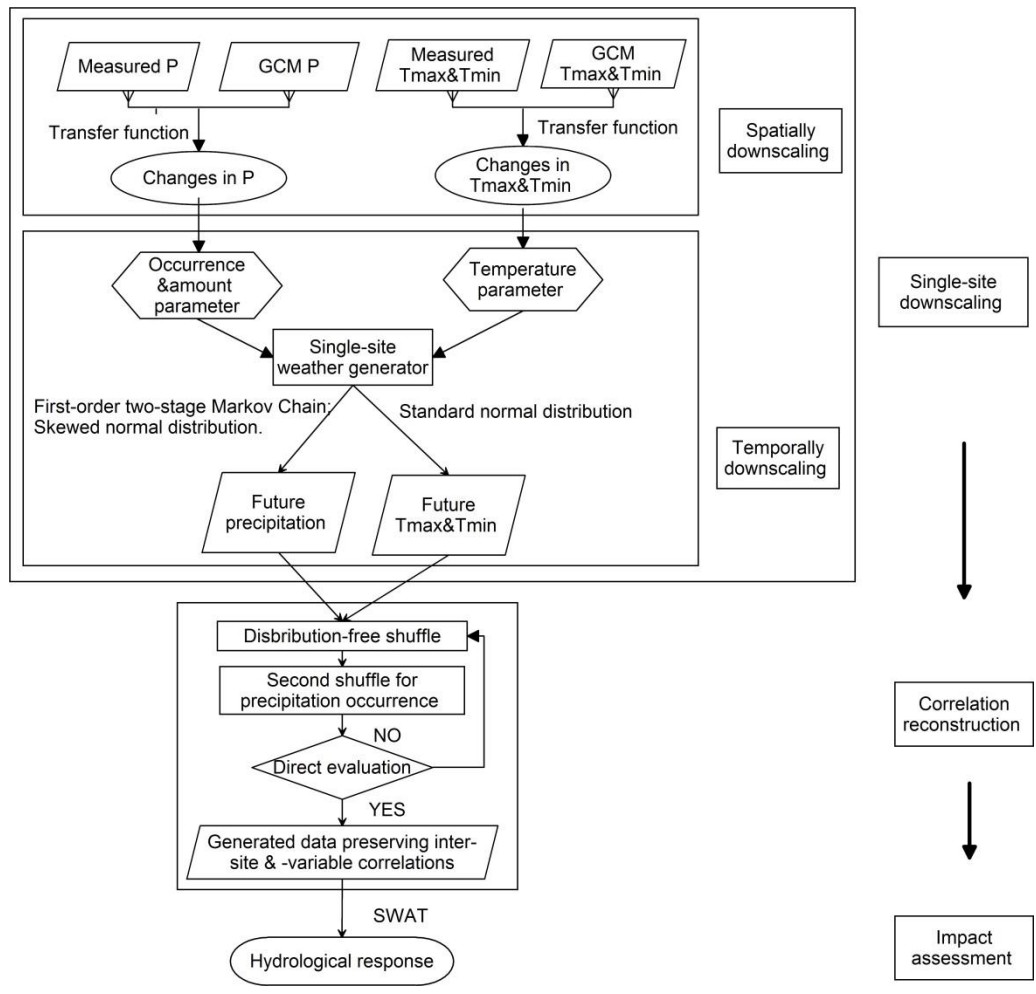

Figure 2. Schematic diagram of the multisite and multivariate GCM downscaling and the structure of this study



For the third step, the method for pairing independent variables to induce the desired rank correlations (Iman and Conover, 1982), which was successfully used in our previous study for multisite precipitation simulation by the two-stage weather generator (TSWG) (Li, 2014), was introduced to obtain the multisite and multivariate correlations. The theoretical basis is described as follows. To assign a desired correlation matrix $[C]$ to a random row vector $[X]$, two steps should be carried out.

$[C]$ is firstly decomposed to $[C]=[R][R']$, and then the upper triangular matrix $[R']$ is used to multiply $[X]$ to result in a new matrix $[X][R']$ with the desired correlation matrix $[C]$. Specifically, for this study, the daily time series of P, $T_{max}$, and $T_{min}$ generated by SSWG for all stations and for each month were put in one matrix, where the rows represented days and the columns represented stations and variables. Then the ranks of each column were converted to a standard normal distribution by calculating the van der Waerden scores, which can be calculated by $\Phi^{-1}\{i/(n+1)\}$, where $\Phi^{-1}$ is the inverse function of

the standard normal distribution, and $i$ stands for the rank of each column. The score matrix was multiplied by the decomposed correlation matrix $[R']$ to get a new matrix with the target correlation coefficients. The ranks in the new matrix were used to shuffle the raw matrix. During the above shuffle procedure, the non-positive correlation matrices and the tied ranks due to dry days should be adjusted. The non-positive correlation matrices were amended by a spectral decomposition procedure (Rebonato and Jäckel, 2000). The tied ranks can be solved by assigning small values of less than the threshold

definition of a wet event (0.1 mm in this study) to dry days. As the data rearrangements perturb the occurrence structure, the occurrence adjustment should be carried out according to those from SSWG. Detailed procedures can be found in Li (2014).

## 2.3 Hydrological simulation

SWAT, a physically based distributed hydrological model for studying the impact of environmental changes on hydrology (Arnold et al., 1998), was employed to evaluate the response of streamflow to climate change. SWAT was calibrated using

the observed data, and then it was used to simulate hydrological processes for the period of 2011–2040. The two main components in the hydrological cycle, i.e. runoff and potential evapotranspiration ($ET_0$), were respectively simulated with the curve number method (USDA-SCS, 1972) and the Hargreaves method (Hargreaves et al., 1985). The Hargreaves method was chosen out of two other options (Penman-Monteith and Priestley-Taylor) because the climate projection in this study only considered changes in temperature. As the Hargreaves-based $ET_0$ depends mostly on temperature, the changes in

projected temperature can thus be better translated into evaporation losses.

Monthly streamflow for the period of 1960–1970 in Zhangjiashan gauge station, the watershed outlet, was used to calibrate SWAT. The period of 1960–1970 was chosen because of the small intensity of human activities and changes in climate. The soil conservation measures as well as the other human activities were the smallest before 1970 and thus have the minimum impacts on rainfall-runoff relationships. The period before 1970 is therefore used by the Yellow River Conservancy

Commission as the baseline to assess the effects of soil conservation measures. Therefore, 1961–1970 is a reasonable period for SWAT calibration and validation.





1960–1964 was used for model calibration and 1965–1970 was used for model validation where the calibrated parameters were used. Automated calibration/validation and uncertainty analysis were carried out by the Sequential Uncertainty Fitting version 2 (SUFI-2) in SWAT-CUP (Abbaspour et al., 2007). After sensitivity analysis, the parameters most responsible for runoff simulation were identified and calibrated with the objective function of the Nash-Sutcliffe efficiency coefficient. In

this study, the sensitive parameters mainly include curve number (CN), base flow recession coefficient (ALPHA_BF), soil evaporation coefficient (ESCO), available water capacity (SOL_AWC), and groundwater delay time (GW_DELAY). The Nash-Sutcliffe efficiency coefficients for the two periods were both 0.78, indicating that SWAT can satisfactorily simulate the streamflow (Figure 3).

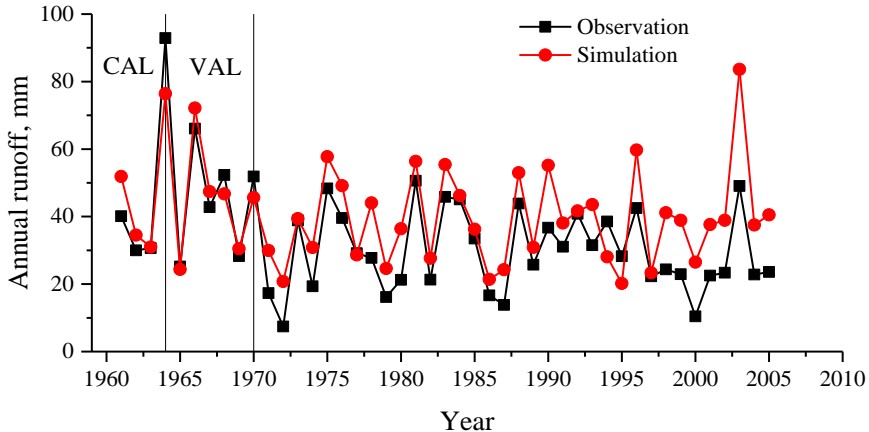

Figure 3. Observed and simulated runoff in Zhangjiashan station of the Jing River during 1961-2005 (CAL and VAL shows the period for model calibration and validation)

Although significant soil and water conservation projects in our study basin since the 1990s have affected the runoff processes and streamflow amount, they were not taken into account in our SWAT simulations, which may cause simulation errors for the period of 1991–2005. To exclude the impacts of human activities, the natural runoff represented by the SWAT-

simulated runoff for the period of 1960–2005 was used hereafter as baseline to emphasize the climate-induced changes in runoff.

During the hydrological simulation for both current and future periods, the land surface conditions are assumed to be invariant. According to our analysis, the land use pattern in 2010 is similar as that in 1986 though there is variability during 1986-2010 (Li et al., 2016). Further, the current vegetation is approaching the sustainable water resource limits (Feng et al.,

2016). Thus, the vegetation cannot increase in future and probably keep invariant according to the land use planning of the local government. Accordingly, the assumption of the invariant land use pattern is reasonable.



## 2.4 Statistical analysis

To draw reliable conclusions regarding changes in climate and hydrology, the significance of future changes from the baseline was examined with the student's $t$-test (p=0.05). Univariate linear regression analysis was used to obtain the sensitivity coefficients between climate changes and streamflow response. Specifically, based on the data of 18 scenarios×12 months, the changes in mean and variance of monthly P, $T_{max}$, and $T_{min}$ were used to develop relationships with the changes in the mean, variance, or extremes of monthly streamflow, respectively. Then the slope of the linear equation was used as the sensitivity coefficient.

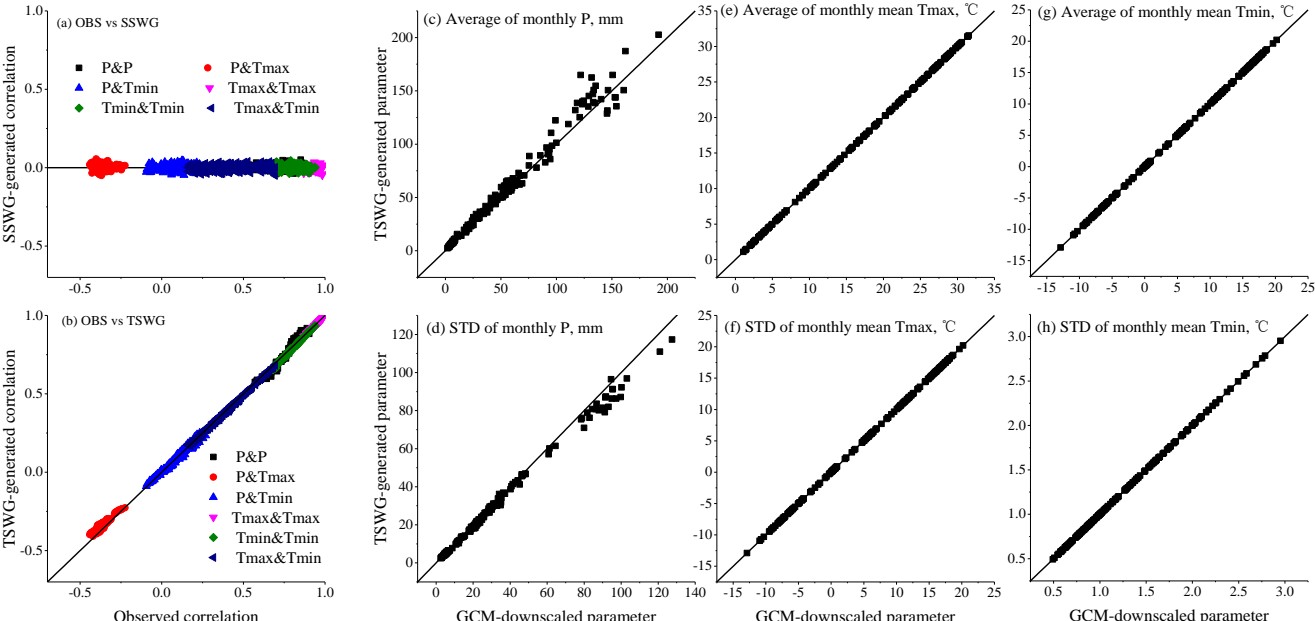

Figure 4. a-b, observed (OBS) versus generated multisite and multivariate correlations of daily precipitation (P), maximum ($T_{max}$) and minimum temperature ($T_{min}$); c-h, average and standard deviations (STD) of monthly P, $T_{max}$ and $T_{min}$

## 3 Results and Discussion

### 3.1 Performance of correlation reconstruction method

To validate the performance of the proposed method, the downscaled parameters related to changes in precipitation and temperature from RCP2.6 in CanESM2 were used to generate multisite and multivariate climate change scenarios. Since no correlation was considered during the single-site GCM downscaling, the inter-site and -variable correlations fluctuated around zero (Figure 4a). However, after rearranging the structure of data matrix, the multisite and multivariate correlations were well reproduced (Figure 4b). In addition, the average and standard deviations of monthly precipitation and monthly mean temperature were well reproduced since the shuffle procedure did not change them from SSWG (Figure 4c-h). The





slightly underestimated standard deviations of monthly precipitation were caused by the inherent weakness of weather generator, i.e. underestimation of low-frequency variability. The statistics of monthly mean temperature were almost the same as the GCM-downscaled parameters. The above results imply that the proposed method is effective in correlation reconstruction, and can satisfactorily reproduce the statistics including low-frequency variability.

To carry out an impact assessment of climate change, the projected climate changes should be transferred to hydrological simulation to the greatest extent. The observed precipitation and temperature for the period of 1961–2005 were used to generate a 100-year climate with SSWG and TSWG to drive SWAT, and then the simulated hydrological statistics were compared with observations to ensure the correlation reconstruction method did not bring errors (Figure 5). Obviously, the three series of climate gave a similar monthly mean streamflow; SSWG underestimated the variances and maxima of the
monthly streamflow while overestimated its minima; however, TSWG gave similar variances and extremes of the monthly streamflow except for a few months. The above results suggested that the developed multisite and multivariate climate change scenarios based on TSWG can be effectively used to simulate hydrological variability and extreme.

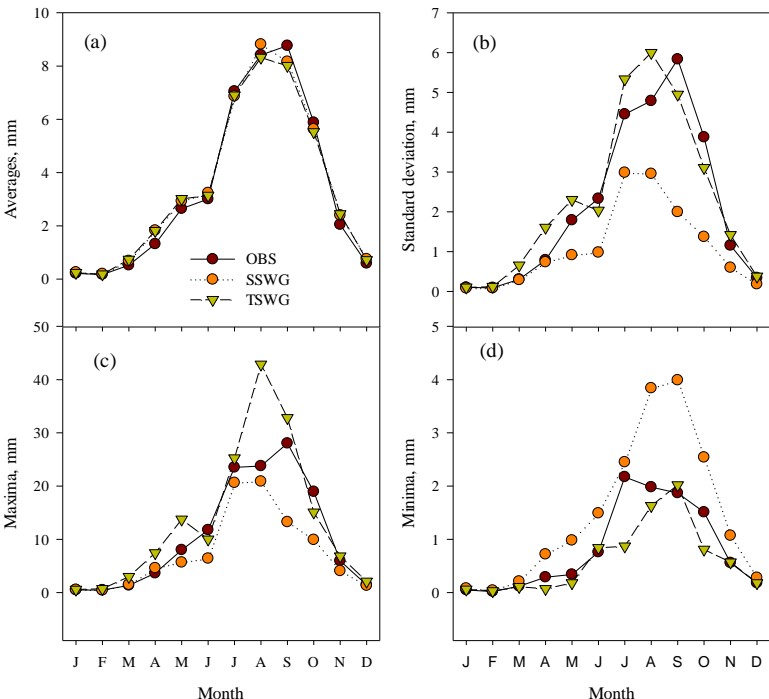

Figure 5. Statistics of monthly streamflow for the observed (OBS) and simulated by SSWG- and TSWG-generated climate.
The observation is for the period 1961–1990, the simulated runoff is from the 100-year climate generated from the statistical parameters from the observed climate.



## 3.2 Projected climate changes

The climate over 2011–2040 compared to that over 1961–2005 appeared to be drier and warmer under most scenarios (Table 2), and the trend was more significant under higher RCPs. Averaged over all scenarios, the annual mean precipitation decreased by –1.3±4.4%, while $T_{max}$ and $T_{min}$ increased by 1.6±0.3 and 1.3±0.2 ℃, respectively. Comparing the projected climate from all scenarios with the baseline data (p=0.05), the annual mean precipitation would not change, while temperature would significantly increase during 2011–2040.

Table 2. Relative changes in annual mean climate between 1961-2005 and 2011-2040

| RCP | P Change, % | | | | $T_{max}$ Change, ℃ | | | | $T_{min}$ Change, ℃ | | | |
|---|---|---|---|---|---|---|---|---|---|---|---|---|
| | 2.6 | 4.5 | 6.0 | 8.5 | 2.6 | 4.5 | 6.0 | 8.5 | 2.6 | 4.5 | 6.0 | 8.5 |
| CanESM2 | +9.2 | -4.2 | — | +5.1 | +1.8 | +1.7 | — | +2.0 | +1.4 | +1.4 | — | +1.6 |
| CSIRO_3.6.0 | +0.3 | +0.3 | -3.3 | -3.3 | +1.8 | +1.4 | +1.1 | +1.6 | +1.5 | +1.2 | +0.8 | +1.3 |
| GFDL_CM3 | +0.1 | -1.5 | +1.3 | -7.7 | +1.8 | +1.8 | +1.5 | +2.1 | +1.4 | +1.4 | +1.2 | +1.6 |
| HadGEM2-ES | -6.3 | -2.2 | -2.9 | -6.7 | +2.0 | +1.6 | +1.5 | +2.0 | +1.5 | +1.3 | +1.2 | +1.4 |
| MPI-ESM-LR | +4.5 | -0.7 | — | -4.7 | +1.2 | +1.2 | — | +1.3 | +1.2 | +1.2 | — | +1.3 |
| Mean-each RCP | +1.5 | -1.7 | -1.6 | -3.4 | +1.7 | +1.6 | +1.4 | +1.8 | +1.4 | +1.3 | +1.1 | +1.4 |
| p-each RCP | 0.29 | **0.05** | 0.19 | 0.10 | **<0.01** | **<0.01** | 0.01 | **<0.01** | **<0.01** | **<0.01** | **<0.01** | **<0.01** |
| Mean-all RCPs | -1.3 | | | | +1.6 | | | | +1.3 | | | |
| p-all RCPs | 0.12 | | | | **<0.01** | | | | **<0.01** | | | |

Mean-each/all RCP, average changes for all GCMs under one/all RCP;

p-each/all RCP, significance of t-test for all GCMs under one/all RCP.

The differences in monthly mean climate between 2011–2040 and 1961–2005 showed the seasonal patterns of climate change (Figure 6). Precipitation significantly decreased from August to October while increasing from November through March and in May (Figure 6a), and temperature increased significantly across all seasons (Figure 6b and c). Therefore, a drier climate would be expected during the flood season while a wetter climate might exist for winter and spring during 2011–2040.

The variances in precipitation and temperature during 2011–2040 relative to 1961–2005 tended to increase under most scenarios (Table 3). Averaged over all scenarios, the variance ratios of P, $T_{max}$ and $T_{min}$ were 1.15±0.13, 1.15±0.14 and 1.04±0.10, respectively. The significance test (p=0.05) further confirmed the variance increase for P and $T_{max}$, which suggested that future climate would be more variable than the present climate.

The variance of monthly precipitation tended to increase under most scenarios and for most months (Figure 6d); however, the upward trends in variances were only significant for six months ($1^{st}$, $3^{rd}$, $5^{th}$, $7^{th}$, $11^{th}$ and $12^{th}$ month). For temperature, monthly variances increased over the first half of the year and decreased in the second half of the year (Figure 6e&f), and the





significance test further showed that $T_{max}$ variances significantly increased in the first half of the year except for March and May, while $T_{min}$ variances significantly increase from March to May. Overall, the increase in climate variability is significant during the first half of the year.

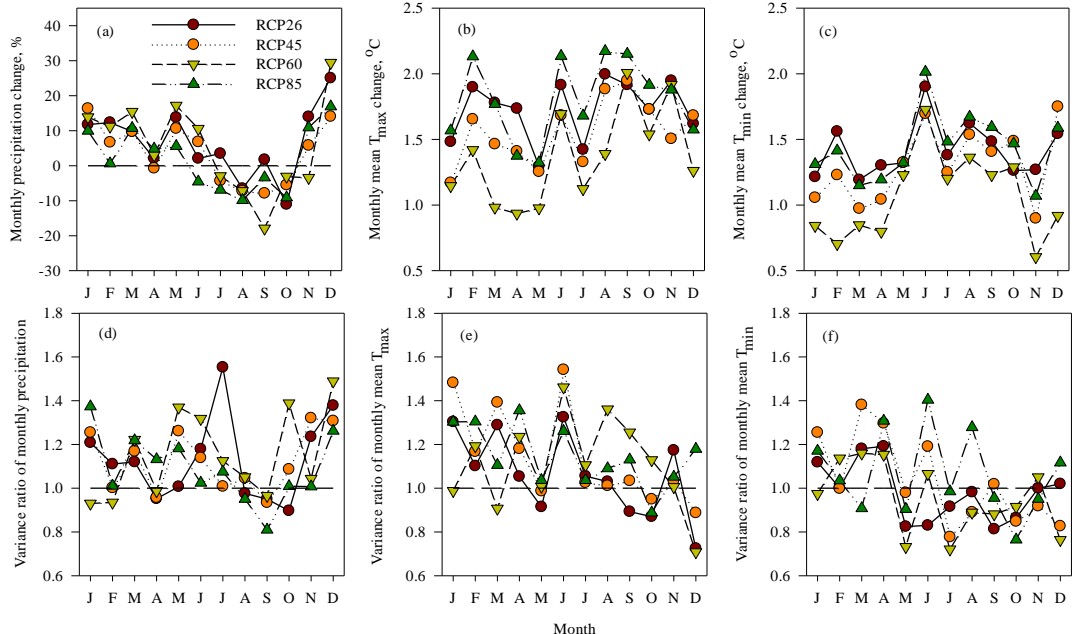

5        Figure 6. Changes in the averages and variances of monthly climate during 2011–2040 relative to 1961–2005

Table 3. Variance ratios of precipitation and temperature during 2011-2040 relative to 1961-2005

| RCP | P variance ratio | | | | $T_{max}$ variance ratio | | | | $T_{min}$ variance ratio | | | |
|---|---|---|---|---|---|---|---|---|---|---|---|---|
| | 2.6 | 4.5 | 6.0 | 8.5 | 2.6 | 4.5 | 6.0 | 8.5 | 2.6 | 4.5 | 6.0 | 8.5 |
| CanESM2 | 1.34 | 1.12 | — | 1.19 | 1.18 | 1.41 | — | 1.40 | 1.00 | 1.27 | — | 1.20 |
| CSIRO_3.6.0 | 1.02 | 1.26 | 1.26 | 1.12 | 1.14 | 1.15 | 1.19 | 1.22 | 0.91 | 1.06 | 0.97 | 1.17 |
| GFDL_CM3 | 1.38 | 1.31 | 1.24 | 1.15 | 1.21 | 1.14 | 1.03 | 1.04 | 1.08 | 1.03 | 0.92 | 1.04 |
| HadGEM2-ES | 1.05 | 1.16 | 1.01 | 1.03 | 1.00 | 1.16 | 1.21 | 1.35 | 1.07 | 0.94 | 1.01 | 1.10 |
| MPI-ESM-LR | 1.00 | 0.93 | — | 1.09 | 0.94 | 1.04 | — | 0.91 | 0.94 | 1.01 | — | 0.95 |
| Mean-each RCP | 1.16 | 1.16 | 1.17 | 1.12 | 1.09 | 1.18 | 1.14 | 1.18 | 1.00 | 1.07 | 0.97 | 1.09 |
| p-each RCP | 0.06 | **0.040** | 0.08 | **<0.01** | 0.07 | **0.02** | 0.06 | 0.06 | 0.50 | 0.16 | 0.16 | 0.06 |
| Mean-all RCPs | 1.15 | | | | 1.15 | | | | 1.04 | | | |
| p-all RCPs | **<0.01** | | | | **<0.01** | | | | **0.07** | | | |

Mean-each/all RCP, average changes for all GCMs under one/all RCP;

p-each/all RCP, significance of t-test for all GCMs under one/all RCP.





### 3.3 Hydrological response to climate changes

The streamflow in the Jing River was simulated with SWAT using the 18 climate change scenarios as external forcing (Table 4). Averaged over all scenarios, the annual mean streamflow decreased insignificantly by 1.0±15%. The seasonal patterns of streamflow changes were similar under all the RCPs (Figure 7a). Monthly mean streamflow decreased from September through November and increased during the other months, and the greatest increase occurred during winter and spring. The *t*-test further showed that the upward trend from November through June and the downward trend in September and October were significant, while no significant trend was detected for July and August.

Table 4. Changes in average and variance of annual streamflow, mean monthly extreme streamflow during the 2020s relative to 1961-2005

| RCP | Changes in average, % | | | | Variance ratio | | | | Changes in maxima, % | | | | Changes in minima, % | | | |
|---|---|---|---|---|---|---|---|---|---|---|---|---|---|---|---|---|
| | 2.6 | 4.5 | 6.0 | 8.5 | 2.6 | 4.5 | 6.0 | 8.5 | 2.6 | 4.5 | 6.0 | 8.5 | 2.6 | 4.5 | 6.0 | 8.5 |
| CanESM2 | 38 | -8 | — | 2 | 2.85 | 1.23 | | 0.98 | 75 | 31 | — | 27 | -10 | -25 | — | -18 |
| CSIRO-3.6.0 | -6 | 7 | 2 | -6 | 0.76 | 1.32 | 1.10 | 0.99 | 9 | 64 | 65 | 15 | -34 | -18 | -45 | -34 |
| GFDL-CM3 | 9 | 5 | 3 | -17 | 1.32 | 1.70 | 1.13 | 0.75 | 43 | 43 | 34 | 25 | -32 | -33 | -39 | -49 |
| HadGEM2-ES | -16 | -10 | 26 | -6 | 0.71 | 1.05 | 2.25 | 1.07 | -5 | 21 | 60 | 22 | -36 | -40 | -29 | -47 |
| MPI-ESM-LR | 16 | -3 | — | -17 | 1.51 | 1.15 | | 0.68 | 42 | 21 | — | 9 | -35 | -20 | — | -46 |
| Mean-each RCP | 8 | -2 | 10 | -9 | 1.43 | 1.29 | 1.49 | 0.89 | 33 | 36 | 53 | 20 | -30 | -27 | -37 | -39 |
| p-each RCP | 0.22 | 0.31 | 0.16 | **0.03** | 0.16 | **0.03** | 0.17 | 0.17 | | **<0.01** | | | | **<0.01** | | |
| Mean-all RCPs | 1.0 | | | | 1.25 | | | | 33 | | | | -33 | | | |
| p-all RCPs | 0.39 | | | | **0.03** | | | | **<0.01** | | | | **<0.01** | | | |

Mean-each/all RCP, average changes for all GCMs under one/all RCP;

p-each/all RCP, significance of t-test for all GCMs under one/all RCP.

The variances of annual streamflow during 2011–2040 relative to 1961–2005 increased under most scenarios (Table 4). Averaged over all scenarios, the variance ratios of annual streamflow were 1.25±0.55 (p=0.03), which implies that the inter-annual variability of future streamflow would be more significant. The variances of monthly streamflow had similar seasonal patterns under all RCPs (Figure 7b), and significantly increased from November through August except for January and June while decreased in October (p=0.05), which implies that intra-annual variability would also be greater during 2011–2040.

The maximum/minimum monthly streamflow increased/decreased significantly by 33±22% and -33±11%, respectively (Table 5). The monthly maxima increased for most months except October (Figure 7c, and October was excluded by the *t*-test). The monthly minima decreased for most months except for an increase in January and February (Figure 7d), and the downward trend in April and from June through November and the upward trend in January, February and May were confirmed by *t*-test. The combined effects of the upward trend in the maxima and the downward trend in the minima led to the increase in variability.



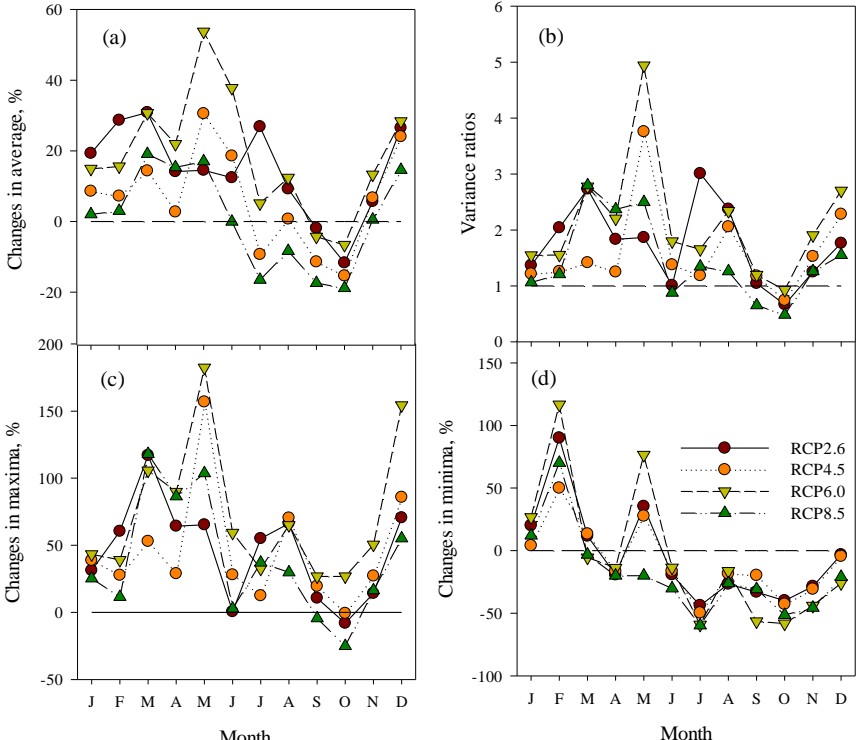

Figure 7. Changes in average and variability of monthly streamflow during 2011–2040 relative to 1961–2005

## 3.4 Links between climate change and streamflow variability

According to the sensitivity analysis, precipitation changes during 2011–2040 had a significant positive correlation with streamflow changes (p<0.01) (Table 5). The precipitation changes had smaller impacts on the mean streamflow than on the streamflow variances; a 1% increase in precipitation means or variances increased mean streamflow by 0.3%–0.6%, while it increased streamflow variances by 1.5%–1.9%. Changes in either the means or variances of precipitation had similar impacts on hydrological extremes; a 1% increase in precipitation increased extreme monthly streamflow by about 1%.

Changes in temperature means negatively correlated with streamflow means and variances, while changes in temperature variances positively correlated with streamflow variances and extremes (Table 5). The impacts of changes in temperature variances were much smaller than those of temperature means. A 1 ℃ increase in $T_{max}$ significantly decreased mean streamflow by 7.9% (p<0.01), while it decreased streamflow variance by 24.8% (p=0.04).

Overall, the main factors controlling streamflow variations were the changes in P variances and mean $T_{max}$ (Table 5). For example, changes in precipitation means and variances of –1.3% and +15% altered mean streamflow by –0.8% and +9.0%, while changing streamflow variances by –2.5% and +30.0%, respectively. Therefore, streamflow was more sensitive to





changes in precipitation variances and temperature means, while it is less susceptible to changes in precipitation means and temperature variances.

Table 5. Sensitivity of streamflow to the changes in mean and variances of precipitation and temperature

| | $P\&R_m$, $P\&R_v$ | | | | $TX\&R_m$, $TX\&R_v$ | | | | $TN\&R_m$, $TN\&R_v$ | | | |
|---|---|---|---|---|---|---|---|---|---|---|---|---|
| | $P_m\&R_m$ | $P_v\&R_m$ | $P_m\&R_v$ | $P_v\&R_v$ | $TX_m\&R_m$ | $TX_v\&R_m$ | $TX_m\&R_v$ | $TX_v\&R_v$ | $TN_m\&R_m$ | $TN_v\&R_m$ | $TN_m\&R_v$ | $TN_v\&R_v$ |
| k | +0.6 | +0.3 | +1.9 | +1.5 | -7.9 | +0.1 | -24.8 | +0.1 | -2.7 | +0.04 | -15.8 | -0.03 |
| p | <0.01 | <0.01 | <0.01 | <0.01 | <0.01 | 0.09 | 0.06 | 0.47 | 0.44 | 0.40 | 0.38 | 0.90 |
| | $P\&R_x$, $P\&R_n$ | | | | $TX\&R_x$, $TN\&R_X$ | | | | $TX\&R_n$, $TN\&Rn$ | | | |
| | $P_m\&R_x$ | $P_v\&R_x$ | $P_m\&R_n$ | $P_v\&R_n$ | $P_m\&R_x$ | $P_v\&R_x$ | $P_m\&R_n$ | $P_v\&R_n$ | $P_m\&R_x$ | $P_v\&R_x$ | $P_m\&R_n$ | $P_v\&R_n$ |
| k | +1.0 | +0.6 | +0.9 | +0.1 | -10.9 | +0.1 | -7.4 | +0.1 | -11.5 | +0.1 | -5.7 | +0.1 |
| p | <0.01 | <0.01 | <0.01 | <0.01 | 0.10 | 0.18 | 0.41 | 0.57 | 0.02 | 0.33 | 0.38 | 0.24 |

$R_m$, $R_v$, $R_x$, $R_n$—the changes in mean (%), variance (%), maximum (%), and minimum (%) monthly streamflow;

$TX_m$, $TX_v$, $TN_m$, $TN_v$—the changes in mean (℃) and variance (%) of $T_{max}$ and $T_{min}$;

k, p—the slope and significance level of linear regression.

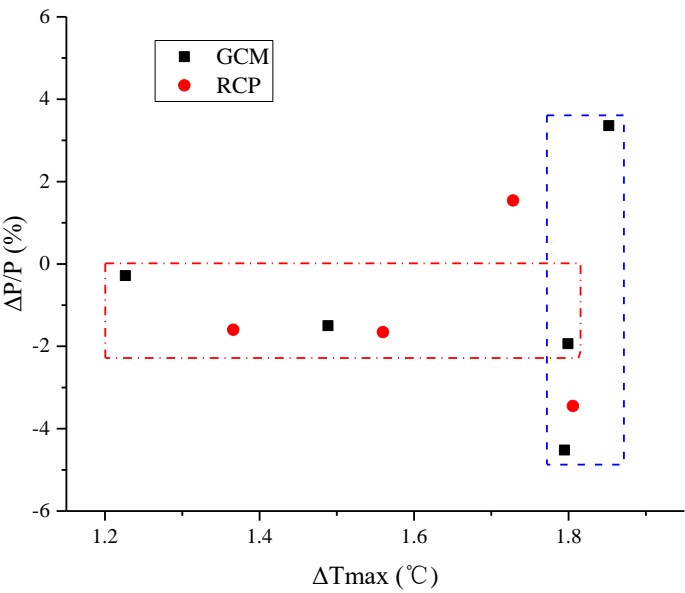

Figure 8. Projected changes (Δ) in annual mean precipitation (P) and maximum temperature ($T_{max}$) averaged over GCMs or RCPs

## 5  3.5 Uncertainties in impact studies

For climate change impacts on hydrology, uncertainties are usually from GCM, emission scenarios, downscaling and hydrological modeling. However, in most cases, the uncertainties from climate model structure are greater than those associated with hydrological model or downscaling method (Wilby and Harris, 2006; Arnell, 2011; Chen et al., 2011; Gosling et al., 2011). For this study, as the climatic statistics and multi-site multivariate correlations were well reproduced




and SWAT was calibrated fairly well, the uncertainties from them were small. However, the uncertainties from GCMs and RCPs were much greater. Averaged over all GCMs and RCPs, annual mean precipitation would change by -1.3±4.4% (Table 2), and the large uncertainty in climate projection caused an even larger uncertainty in runoff change. Specifically, the annual mean runoff changed by 1±15%, while the monthly maximum and minimum streamflow respectively changed by 33±22% and -33±11% (Table 4).

Averaged over all projections over a certain GCM or RCP, the uncertainty related to GCMs and RCPs was large (Figure 8). For example, three GCMs (in the horizontal dash-dotted red rectangle in Figure 8) projected similar changes in precipitation, but different changes in temperature; three GCMs (in the vertical dashed blue rectangle in Figure 8) projected similar temperature changes but quite different precipitation changes.

The sensitivity of the uncertainties linked to GCMs was reduced in the hydrological response (Figure 9), where four out of five GCMs projected similar changes in the mean or variance for runoff. Figure 9 showed that CanESM2 model contributed greatly to the uncertainty envelope of runoff changes. Different from GCMs, the sensitivity of the uncertainties related to RCPs increased in impact assessment, where the changes in annual mean streamflow covered greater absolute ranges than those of precipitation and temperature. The projected change directions in streamflow were even different for different RCPs.

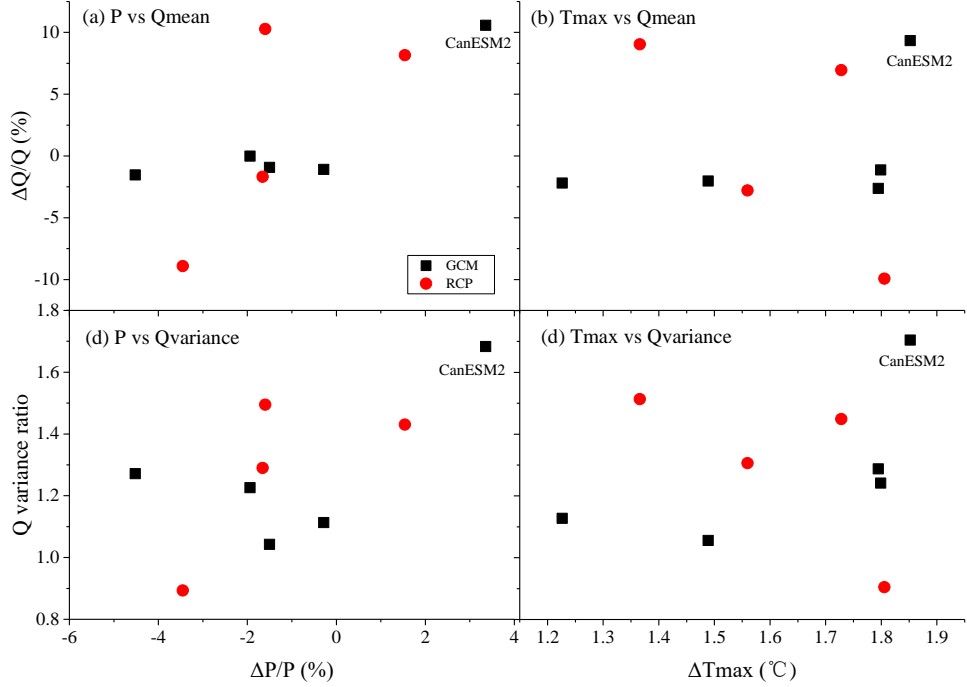

Figure 9. Projected changes (Δ) in precipitation (P), maximum temperature ($T_{max}$) and streamflow (Q) averaged over GCMs or RCPs





## 4 Conclusions

To better simulate the hydrological variability with distributed hydrological models, an efficient multisite and multivariate GCM downscaling method was presented to generate climate change scenarios. Compared with those needing considerable parameter estimation and statistical verification, this method only uses climatic statistics and spatiotemporal correlations without excessive iterations. The preprocessing multisite weather generator reconstructs the correlations for each variable and for each process, but this method can be applied to all variables and all processes at the same time. As a postprocessing method, it is applicable to the single-site climate scenarios generated by any algorithm, such as the chain-dependent process of the Richardson-type method (Richardson, 1981), the empirical method of LARS-WG (Semenov and Barrow, 1997), or the circulation-based weather generator of SDSM (Wilby et al., 2002).

Based on the developed climate change scenarios and hydrological modeling in the Jing River Catchment on China's Loess Plateau, both annual mean precipitation and streamflow would not change, which implies that the gross amount of water resources might be similar to that of the present time. Further analysis found that the precipitation variances had greater impacts than precipitation means on the streamflow, and the high flows tended to be sensitive to precipitation changes while the low flows were determined by temperature changes. According to the results of this study, the rising variance in streamflow suggests more floods during the summer and severe water shortage during the winter and spring, which implies that the available water resources could possibly decrease and the soil erosion could be more severe. Although human activities might reduce flooding to some extent, we should still pay great attention to the potential changes in extreme hydrological events.

The changes in the statistical parameters of precipitation and/or temperature, such as mean and variances, played different role in the mean state and variability of streamflow. The changes in precipitation/temperature were respectively positively/negatively correlated with those in streamflow. Specifically, the increase in streamflow variances was mainly attributed to the increase in precipitation variances and temperature means, and the positive contribution from increased precipitation variances was larger than the negative contributions from increased temperature means.

The method for multisite and multivariate correlation reconstruction slightly perturbed the precipitation occurrence (Li, 2014), which is the inherent weakness of the postprocessing methods (Clark et al., 2004a; Bárdossy and Pegram, 2012). So far, it can be satisfactorily used for streamflow modeling on monthly scale in a large catchment with an area of 45,421 km$^2$ and 15 weather stations, which is good enough for sensitivity analysis or impact assessment. To further validate its applicability on different spatial and temporal scales, the method should be applied to daily streamflow simulation and some other watersheds with different areas and climate.





**Conflict of interest**

The authors declare that there is no conflict of interest regarding the publication of this paper

**Acknowledgments**

This study is jointly funded by the National Natural Science Foundation of China (41101022), Fok Ying Tung Foundation
(141016), and Chinese Universities Scientific Fund (2452015105). We appreciate the Data Sharing Infrastructure of Loess
Plateau for providing streamflow and climate data. We acknowledge the World Climate Research Programme's Working
Group on Coupled Modelling, which is responsible for CMIP, and we thank the climate modeling groups (listed in Table 1)
for producing and making available their model output.

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
