# Peer review of "Evaluating climate change impacts on streamflow variability based on a multisite multivariate GCM downscaling method"

_Hydrology and Earth System Sciences, 2017_

## Short Comment (SC1) · 23 Jun 2017

The authors attempt to investigate the impacts of climate change on streamflow for the Jing River catchment on China's Loess Plateau. To achieve this objective, a statistical downscaling approach based on a transfer function and a stochastic weather generator is employed for downscaling daily precipitation and temperatures from five GCMs under four IPCC 2013 scenarios. The downscaling is performed at 18 meteorological stations. For generating streamflow, the downscaled variables are used as forcing data for a hydrological model (SWAT). Climate change signals for streamflow are evaluated for the period 2011–2040 relative to 1961–2005. The authors found that streamflow

variability would be greater over most months at a seasonal scale due to the increase of monthly maximum streamflow and the decrease of monthly minimum streamflow. The increase in streamflow variability was mainly attributed to the larger positive contributions from increased precipitation variances than the negative contributions from increased temperature means.

General comments

The multi-site multivariate downscaling approach employed by the authors is promising especially its distribution-free attribute which shows performance close to existing parametric weather generators. Its ability to reproduce observed spatial and intervariable dependency structures is very attractive especially for flood adaptation studies.

Specific comments

- The authors claim that the study is restricted to the period 2011-2040 for purposes of near-term adaption planning. The study is carried out for 4 scenarios. However, there is really no difference in scenarios before 2050 after which they start to diverge. So I doubt how the authors used 4 scenarios for which the results would just look alike. - Please include a paragraph at the end of the discussion which clearly states the purpose of this study and whether previous studies have been performed in the same study area, what were the limitations and why your study adds value to existing ones? - What is the rationale for selecting the 4 ESMs used in this study? - The statistical performance test plots in Figure 4 look too good. Are the weather gauges in the study area very homogenous with less inter-station differences? It could be that the stations are very similar synoptically, thus it is easy to reproduce the observed weather. - Figure 5 shows that the weather generators could reproduce just the mean of streamflow but the variances and extremes were not well reproduced, something that is problematic for climate change impacts analyses. You would also notice that not just the magnitude is different but also the phasing of streamflow. - Mean-all RCPs in Table 2: what is the purpose of averaging across RCPs? These are very different pathways and needless

averaging them.

Technical corrections

- Change GCMs to ESMs. The CMIP5 products are earth system models because they close the carbon cycle. - Page 5 line 1: climate changes, delete the "s" in changes. - Figure 2: spatial/temporal downscaling and not spatially/temporally downscaling. - Line 1 page 8: don't start a sentence with 1960–1964. Say "The period 1960–1964…"
* * *

---

## Referee Comment (RC2) · Anonymous Referee #2 · 14 Sep 2017

The manuscript titled "Evaluating climate change impacts on streamflow variability based on multisite multivariate GCM downscaling method" applies a multivariate downscaling approach to the Jing River Basin. Overall I found the manuscript to be of sound technical quality and does suggests a novel approach of GCM downscaling as inputs to a hydrologic model to investigate hydrologic variability. However, I believe that this manuscript requires additional work to clarify the text to best present this work. Please see my general and specific comments for additional guidance. I found the title to be accurate for the manuscript's content, but would suggest the addition of the words "in the Jing River Basin" or something equivalent.

General Comments: - As mentioned, I find the manuscript to be of good technical quality and appears to use industry standard model techniques to explore the effects of this novel downscaling approach.

- The figures seem appropriate for the manuscript topic and seem to support the findings stated in the body of the manuscript.

- The first paragraph of the introduction seems to be the weakest of the introduction. The english/grammar should be improved for clarity. My opinion is that the first paragraph should be reworked. The syntax/grammar of the remainder of the manuscript should also be refined before publication, but there are no specific sections to highlight other than the first paragraph of the introduction.

- Because this work is based on techniques that the author has already published, I found it difficult in to decipher the contributions of this paper from the previous downscaling work. I highly suggest that the authors clearly present the objectives of the manuscript at the end of the manuscript and how this is differentiated from previous work. What are the primary hypotheses of the work? Whether or not these hypotheses are supported should be contained within the discussion section.

- As it stands, I found the results/discussion section to read like a results section with a very brief discussion at the end of this section. Normally a results/discussion section is needed only when it is difficult to present the results of the paper apart from the greater context of how the results relate to other bodies of work. In this result/discussion section, I found only one location where references are presented, which I found to be insufficient. The most important contribution of this paper will lie on how these results relate to a vast collection of other work completed before this one and what we learned in this paper that can help inform future papers. The core of this manuscript seems to be the attempt to investigate the characterization of hydrologic variability, which has a very long history and should shown in the context of such. It is my opinion that the results/discussion section should be split into separate results and discussion sec-

tions, and therefore the authors should provide an indepth discussion section currently absent from the manuscript.

- The conclusion section should never present new information, however it seems that the authors merged part of a would be discussion section into the conclusions. Based on how the discussion section is written, I might also recommend revising the conclusions section to make sure that new information is not first presented there.

Specific Comments: Page 5 Line 7: This sentence is a bit confusing. The phrase "spatially downscaled GCM outputs from the monthly scale to a daily scale" is a bit clumsily worded. I suggest revising for clarity.

Page 5 Line 10: The first step is to spatially downscale and the second is the temporally downscale. However, you seem to combine them both here to speak specifically about single site GCM downscaling. But then in the same sentence, you only mention the temporal downscaling. Please revise this sentence for clarification.

―――――――――――――――――

---

## Author Comment (AC1) · 2 Oct 2017

Dear Zilefac Elvis Asong,

Thank you very much for the comments. Your concerns about the period selection and innovation are helpful to improve this paper. Please check the following response for your questions. We will modify the paper according to these comments.

The authors attempt to investigate the impacts of climate change on streamflow for the Jing River catchment on China's Loess Plateau. To achieve this objective, a statistical downscaling approach based on a transfer function and a stochastic weather generator is employed for downscaling daily precipitation and temperatures from five GCMs under four IPCC 2013 scenarios. The downscaling is performed at 18 meteorological stations. For generating streamflow, the downscaled variables are used as forcing data for a hydrological model (SWAT). Climate change signals for streamflow are evaluated for the period 2011–2040 relative to 1961–2005. The authors found that streamflow variability would be greater over most months at a seasonal scale due to the increase of monthly maximum streamflow and the decrease of monthly minimum streamflow. The increase in streamflow variability was mainly attributed to the larger positive contributions from increased precipitation variances than the negative contributions from increased temperature means.

**General comments**

The multi-site multivariate downscaling approach employed by the authors is promising especially its distribution-free attribute which shows performance close to existing parametric weather generators. Its ability to reproduce observed spatial and intervariable dependency structures is very attractive especially for flood adaptation studies.

We appreciate that you like and approve our methods for multi-site and multivariate downscaling.

**Specific comments**

- The authors claim that the study is restricted to the period 2011-2040 for purposes of near-term adaption planning. The study is carried out for 4 scenarios. However, there is really no difference in scenarios before 2050 after which they start to diverge. So I doubt how the authors used 4 scenarios for which the results would just look alike.

The selection of the period 2011-2040 and the four RCPs is justified as follows:

The period 2011−2040 was chosen for impact study because of two reasons: (i) the results for the near term can be directly applied for adaption, and (ii) the uncertainties in climate projection for the near term can be significantly reduced since those uncertainties would increase over time due to the nature of ESMs and greenhouse gas

emission projections (IPCC, 2013). The following figure is an updated version for near-term projections from Figure 11.25a in the IPCC AR5 report. This figure shows that the ESMs used in the report can generate at least a 1$^{\circ}$C uncertainty range in temperature projections by 2040, which is very remarkable at the global scale. In addition, we can see that the ESM projections used in this study also produce a large uncertainty range of the streamflow projections over the same period as shown in Figure 7. Thus, it is clear that our modelling results are significantly different with the selected study period and different emission scenarios.

[Figure]

Updated version of IPCC AR5 Figure 11.25a, showing observations and the CMIP5 model projections relative to 1986-2005. The black lines represent observational datasets (HadCRUT4.5, Cowtan & Way, NASA GISTEMP, NOAA GlobalTemp, BEST).

- Please include a paragraph at the end of the discussion which clearly states the purpose of this study and whether previous studies have been performed in the same study area, what were the limitations and why your study adds value to existing ones?

We revised the discussion section to emphasize the importance of this study and its limitations. Extreme hydrological events such as flooding and drought are very important to a region's well-being especially for a watershed with dense population and fragile environments such as our study region. In this study, we produced near-term projections of the streamflow variability at a watershed scale through a downscaling approach that has been fully evaluated in our previous studies and this manuscript, which improved the reliability of our results. In addition, the uncertainty range of our streamflow projections were quantified and well discussed in the manuscript. We believe that all these research results add the values to the existing literature and give strong insights into water resources management especially for extreme hydrological events.

We also discussed the limitations of our results where all our simulations were driven

by the ESM data. Using such an offline approach, the terrestrial feedbacks cannot be taken into account. However, in this inland watershed, land-atmosphere interactions are usually very significant, which could affect the streamflow magnitude and its variability. Therefore, there is a need to further evaluate the effects of those interactions on the runoff processes using a coupled high-resolution climate-hydrology model, a research direction we plan to pursue in the near future.

We split the discussion from results and highlighted the importance of this study. The above modification can be found in either the discussion or the conclusion.

- What is the rationale for selecting the 4 ESMs used in this study?

The selection of the ESMs (actually five) is based on recommendations from two studies for evaluations of climate model skills (Chen and Frauenfeld, 2014; Guo et al., 2013). We believe that these ESMs can basically represent the up-to-date science and technology of climate modeling. We added one sentence in the data description to interpret the rationale in this version.

- The statistical performance test plots in Figure 4 look too good. Are the weather gauges in the study area very homogenous with less inter-station differences? It could be that the stations are very similar synoptically, thus it is easy to reproduce the observed weather.

A figure below shows the inter-station correlations of precipitation, maximum and minimum temperatures. The inter-station correlations are quite different since the points span a large range, implying that the weather stations located in the study area are not homogenous. This conclusion is applicable to the other statistics (Figure 4c-h).

The comparison in Figure 4 is actually focused on the climatology. In our statistical downscaling method, the most important step is to correct the climatology in the simulations based on the observed value, making the two datasets very consistent. This is the common practice for almost all statistical downscaling techniques.

[Figure]

The inter-site correlations of precipitation, maximum and minimum temperature for each month

- Figure 5 shows that the weather generators could reproduce just the mean of streamflow but the variances and extremes were not well reproduced, something that is problematic for climate change impacts analyses. You would also notice that not just the magnitude is different but also the phasing of streamflow.

As we discussed previously, the statistical downscaling approach used in this study is to focus on correcting the long term mean of climate, and variance to some extent. We have not seen a statistical approach that could correct the climate variability accurately based on current sciences, which is determined by climate dynamics. However, we still believe that our projections are very important to understanding the impact of climate variability on streamflow. According to our model calibration and validation, the Nash-Sutcliffe efficiency coefficients are 0.78 for historical monthly streamflow. Thus, the SAWT model forced with those downscaled climate data produces reliable future streamflow variability, which is one of the major focuses in this study. In addition, we mainly examined the differences of our streamflow simulations between two 30-year periods, which are largely controlled by the climate means that have the highest credibility in our modeling results.

- Mean-all RCPs in Table 2: what is the purpose of averaging across RCPs? These are very different pathways and needless averaging them.

We averaged all the RCPs to present a value for the mean precipitation or temperature change. Therefore, it is actually not an average for RCPs, but an average for all ESMs and RCPs, which is interpreted the end of the table.

**Technical corrections**

-    Change GCMs to ESMs. The CMIP5 products are earth system models because they close the carbon cycle.

Except for the introduction, we have replaced all the 'GCMs' with 'ESMs'.

-    Page 5 line 1: climate changes, delete the "s" in changes.

Modified as suggested.

-    Figure 2: spatial/temporal downscaling and not spatially/temporally downscaling.

Modified as suggested.

-    Line 1 page 8: don't start a sentence with 1960–1964. Say "The period 1960–1964…"

Modified as suggested.

---

## Author Comment (AC2) · 2 Oct 2017

Dear anonymous referee #2,

Thank you very much for your comments. Please check the following response to your concerns about innovation and structure adjustment. We will modify the paper according to your suggestions.

The manuscript titled "Evaluating climate change impacts on streamflow variability based on multisite multivariate GCM downscaling method" applies a multivariate down- scaling approach to the Jing River Basin. Overall I found the manuscript to be of sound technical quality and does suggests a novel approach of GCM downscaling as inputs to a hydrologic model to investigate hydrologic variability. However, I believe that this manuscript requires additional work to clarify the text to best present this work. Please see my general and specific comments for additional guidance. I found the title to be accurate for the manuscript's content, but would suggest the addition of the words "in the Jing River Basin" or something equivalent.

We added 'for the Jing River Basin in China' to the end of title.

**General Comments:**

- As mentioned, I find the manuscript to be of good technical quality and appears to use industry standard model techniques to explore the effects of this novel downscaling approach.

-       The figures seem appropriate for the manuscript topic and seem to support the findings stated in the body of the manuscript.

Thank you for the positive comments. We are still trying to present a systematic study with good quality.

-       The first paragraph of the introduction seems to be the weakest of the introduction. The english/grammar should be improved for clarity. My opinion is that the first paragraph should be reworked.   The syntax/grammar of the remainder of the manuscript should also be refined before publication, but there are no specific sections to highlight other than the first paragraph of the introduction.

We rewrote the first paragraph to highlight the importance of projecting the potential changes in the hydrological variability.

-       Because this work is based on techniques that the author has already published, I found it difficult in to decipher the contributions of this paper from the previous downscaling work. I highly suggest that the authors clearly present the objectives of the manuscript at the end of the manuscript and how this is differentiated from previous work. What are the primary hypotheses of the work? Whether or not these hypotheses are supported should be contained within the discussion section.

We added one section 'Why is the proposed downscaling method used?' in the discussion section to highlight the advantages of the proposed method, and also to show the difference of this study

from our previous work.

Overall, our previous work published in 2014 is for multisite precipitation generation. The model was proposed for stochastic generation of one variable at multiple sites. This study extended that model to multisite and multivariate version for future climate change scenarios development. More importantly, we evaluated the changes of hydrological variability with this improved methodology, which is closely related to extreme hydrological events such as flooding and drought. These changes are crucial to water and hazard management, which is a main focus in this study.

- As it stands, I found the results/discussion section to read like a results section with a very brief discussion at the end of this section. Normally a results/discussion section is needed only when it is difficult to present the results of the paper apart from the greater context of how the results relate to other bodies of work. In this result/discussion section, I found only one location where references are presented, which I found to be insufficient. The most important contribution of this paper will lie on how these results relate to a vast collection of other work completed before this one and what we learned in this paper that can help inform future papers. The core of this manuscript seems to be the attempt to investigate the characterization of hydrologic variability, which has a very long history and should shown in the context of such. It is my opinion that the results/discussion section should be split into separate results and discussion sections, and therefore the authors should provide an indepth discussion section currently absent from the manuscript.

Thank this reviewer for the suggestion. We have separated the discussion from the results following this suggestion. In this version, we strengthened the discussion section with answering three important questions, which are: (i) Why do we use the ESM downscaling method described in the text? (ii) How does climate change influence the streamflow variability? (iii) What are the uncertainties and limitations of the projected hydrological changes? The answer to the first question is to focus on discussing the advantages of the proposed method and the difference of this study from the previous work. The answer to the second question discusses the physical mechanism of how the streamflow variability is influenced by climate change and identifies the climate variables that contribute to the streamflow changes. The answer to the third question focuses on the limitations of this study and how to improve our modeling results in the future.

- The conclusion section should never present new information, however it seems that the authors merged part of a would be discussion section into the conclusions. Based on how the discussion section is written, I might also recommend revising the conclusions section to make sure that new information is not first presented there.

We agree. This version moved that paragraph with the new information to the discussion section, and we also revised the discussion section.

**Specific Comments:**

Page 5 Line 7: This sentence is a bit confusing. The phrase "spatially downscaled GCM outputs from the monthly scale to a daily scale" is a bit clumsily worded. I suggest revising for clarity.

We modified the sentence "the second step disaggregated the spatially downscaled GCM outputs from a monthly scale to a daily scale" as "the second step further disaggregated the monthly data to a daily weather series".

Page 5 Line 10: The first step is to spatially downscale and the second is the temporally downscale. However, you seem to combine them both here to speak specifically about single site GCM downscaling. But then in the same sentence, you only mention the temporal downscaling. Please revise this sentence for clarification.

The following sentence "The first and second steps are for single-site GCM downscaling, and the popular technique is to combine transfer function method for spatial downscaling with weather generator for temporal downscaling." is revised as "The first and second steps are for single-site ESM downscaling. The popular technique is to use the transfer function method for spatial downscaling, and then employ the weather generator for temporal downscaling."